# Probing the Nano-Assembly Leading to Periodic Gratings in Poly(p-dioxanone)

**DOI:** 10.3390/nano13192665

**Published:** 2023-09-28

**Authors:** Min-Han Hao, Selvaraj Nagarajan, Eamor M. Woo

**Affiliations:** Department of Chemical Engineering, National Cheng Kung University, No. 1, University Road, Tainan 701-01, Taiwan; qwedefr183@gmail.com

**Keywords:** synchrotron microbeam X-ray, WAXD, SAXS, poly(p-dioxanone), cactus-arm-like ring bands, poly(vinyl alcohol), lamellae self-assembly, iridescence

## Abstract

This study used scanning electron microscopy via 3D dissection coupled with synchrotron radiation with microfocal beams of both small-angle X-ray scattering and wide-angle X-ray diffraction to analyze the periodic crystal aggregates of unusual poly(p-dioxanone) (PPDO) dendritic cactus-arm-like ring bands upon crystallization with a diluent poly(vinyl alcohol) (PVA) that is capable of hydrogen bonding interactions with PPDO. Three-dimensional microscopy interior dissection clearly expounds that the banded periodic architectures are packed by alternately normal-oriented flat-on crystals underneath the valley, periodically interfaced/branched with horizontal-oriented edge-on fibrils underneath the ridge. The oblique angles between the valley’s flat-on crystals with the branches are ca. 25–45° (depending on gradient inclines and bending), which is also proved by the azimuthal angle in microbeam X-ray diffraction. The grating-like strut-rib assembly in the PPDO cactus-arm-like ring bands is further proved by novel iridescence tests.

## 1. Introduction

The biodegradability of polymers is commonly based on aliphatic polyesters or poly(ether ester). Naturally, commercial development has led to extensive interest in biodegradable copolyesters or homopolyesters, such as poly(L-lactic acid) (PLLA) [1], poly(ε-caprolactone) (PCL) [2], poly(β-hydroxybutyrate) (PHB) [3], poly(1,4-butylene succinate) (PBSu) [4], etc. Of the biodegradable polymers, poly(glycolic acid) (PGA) [5] and poly(p-dioxanone) (PPDO) [6] have attracted great interest due to their special potential for biomedical applications. Unlike other commodity biodegradable polymers, PPDO has extraordinary biodegradability and biocompatibility, which make it suitable for potential applications as a biomedical material [7]. This is due to the presence of both ester and ether bonds in the main polymer chains, which provide it with both biodegradability and flexibility as a functional material [8]. PPDO, similar to PGA, has been investigated for biomedical uses in artificial tissues, bone fixation, or drug coatings for controlled delivery, as it has excellent biocompatibility characteristics. However, PPDO has certain weaknesses, namely high cost and low mechanical strength. To overcome such drawbacks, various methods have been researched, which include blending, copolymerization, or inclusion of nanoclays to form nanocomposites, as seen in recent studies of PPDO/PLLA [9], PPDO/PGA [10], nanocomposites made of PPDO/POSS [11], PPDO/CaCO3 [12], etc.

Crystallization of PPDO influences its physical and mechanical properties, and is thus a critical issue. In 2001, Sabino et al. [13] investigated the crystalline morphology of neat PPDO to conclude that neat PPDO spherulites exhibited a ring-banded morphology when crystallized at a low to medium Tc near 65−85 °C. As with many other polymers, when increasing the Tc to higher temperatures (>85 °C), the periodic bands in PPDO spherulites finally transform to a coarse texture with no banding. Obviously, the nominal supercooling (ΔT) can be correlated with the changes in systematic morphology, which are also seen in many other polyesters that display periodic rings. Investigators have also attempted synthesis of copolyesters from PPDO with other polyesters. Poly(p-dioxanone-co-butylene succinate) (PPDOBS) was synthesized by copolymerizing PPDO with diethyl succinate and 1,4-butanediol, whose spherulites were straight-stalk dendrites with no periodic rings when crystallized at a specific Tc (64 °C) [14]. In addition to chemical copolymerization, physical blending is an effective approach to modifying the property of PPDO. Due to its long-chain nature, PPDO can be miscible with only very few polymers. In 2010, Zeng et al. [15] reported the miscibility of poly(p-dioxanone)/poly(ethylene succinate) (PPDO/PESu) blends, and demonstrated that simultaneous crystallization of PPDO and PESu occurred. Due to the complexity of both components being crystallizable, the growth fronts of PESu and PPDO spherulites impinged on each other, leading to superposition of the optical retardation intensity (defined as thickness × birefringence) when viewed in POM.

Periodic assembly in polymers has been debated since 1950 [16]. This phenomenon is seen in some polymers, leading some scientists to propose that the periodicity might be associated with chain folding-induced stresses in long-chain polymers. However, this proposition needs to be tested rigorously, as small-molecule compounds such as phthalic acid (PA) [17,18,19], obviously with the absence of chain folding in its crystals, are also known to display similar periodic ring bands in their crystals. Since then, periodic ring bands have been seen in many polymeric systems [13,20,21,22,23,24,25], which have been extensively researched in past decades. Updated interpretations are also extended to a few arylate polymers. It is instructional to note here that periodic assembly is commonly seen in some natural bio-species that are known to display iridescence, such as the wings of butterflies [26,27,28,29,30], and many others not listed here. Such assembly is known to be packed in orderly gratings with microstructures or nanostructures capable of interference with light.

Blending is an effect technique to custom tailor the properties of polymers. PPDO is partially or fully miscible with a few polymers, such as PPDO/PLLA [9], PPDO/PESu (poly(ethylene succinate)) [15], PPDO/PVA (poly(vinyl alcohol)) [31], and PPDO/PVPh (poly(vinyl phenol)) [32]. As seen from the above list, the possibility of compatibility or miscibility of PPDO with other polymers is enhanced if the candidate polymers possess hydrogen bonding to interact with the carbonyl group of PPDO. As seen in earlier studies [33], PPDO can form periodic ring bands with high regularity more easily if it is blended with suitable diluent polymers that are miscible with it. In addition to 3D microscopy dissection evidence, synchrotron radiation X-ray with a microfocal beam is a powerful tool for investigating morphology within micro-domains. This study utilized both small-angle X-ray scattering (SAXS) and wide-angle X-ray diffraction (WAXD) techniques, both equipped with microfocal beams, to probe the assembly of periodic changes in micro-domains. An earlier study by Nagarajan and Woo [34] proved that X-ray diffraction with an ultrathin microbeam was successful in analyzing the long-debated periodic assembly in poly(ethylene adipate) (PEA). Liao et al. [35] used a similar methodology to shed new light on the interior assembly of PHB. This study extended the applicability of this methodology by analyzing the nanocrystal assembly in PPDO leading to periodicity and iridescence.

## 2. Materials and Methods

Poly(p-dioxanone) (PPDO), a semicrystalline biodegradable poly(ether ester), was supplied by Aldrich Chemical Company, Inc. (St. Louis, MO, USA), with M_w_ = 1.5–2.2 dL/g, 0.1% (*w*/*v*) in hexafluoroisopropanol, T_g_ = −9 °C and T_m_ = 105 °C. Poly(vinyl alcohol) (PVA) was purchased from Scientific Polymer Products, Inc. (USA), with T_m_ = 200 °C, T_g_ = 60 °C. For comparison, film specimens of both neat PPDO and PPDO/PVA blends were used in this study, and were both prepared by solution blending with a concentration of 4 wt.%. Neat PPDO (with no PVA) was dissolved into p-dioxane as a good solvent, while the PPDO/PVA blend was dissolved into 1,1,2,2-tetrachloroethane as a co-solvent. The solution (4 wt.% polymers in solvent) was cast on a glass slide or PI film to form a thin cast film, then placed in a vacuum oven to remove the residual solvent. Samples were heated to maximum melting temperature (T_max_) for 2 min on a hotplate to erase the thermal history, then quickly removed from the hotplate to a thermostatted hot stage that was preset at different crystallization temperatures. Fully crystallized samples were then respectively subjected to post thermal treatments for SEM analyses. The solvent p-dioxane was chosen as an ideal etching agent to wash out the amorphous polymer from crystallized specimens and imperfect domains to acquire clearer crystal assembly of the top-surface and interior lamellar arrangement of crystallized PPDO specimens. The composition of PPDO/PVA was controlled at 95/5 by weight, as this gave the most regularity of ring patterns. Specimens were melted at T_max_ = 220 °C for 2 min, then quickly equilibrated at an isothermal Tc until full crystallization. To control the uniform thickness of the films, a top glass was placed on molten polymer films upon crystallization. This ensured the best quality when specimens were viewed in POM.

### Apparatus and Procedures

Polarized-light optical microscopy (POM and OM). Cast films of neat PPDO or PPDO blend systems were first evaluated using polarized-light microscopy (Nikon Optiphot-2, POM) equipped with a Nikon Optiphot-2, POM, (Tokyo, Japan) digital camera, a CCD digital camera (NFX-35), and a microscopic hot stage (Linkam THMS-600 with a T95 temperature programmer, Linkam Scientific, Redhill, UK). A λ-tint plate (530 nm) was used to make contrast interference colors in POM graphs.

High-resolution field-emission scanning electron microscopy (HR-FESEM). Samples were characterized using high-resolution field-emission scanning electron microscopy (Hitachi SU8010, HR-FESEM, Tokyo, Japan)) for the top surface and interior lamellar assembly. After proper fracturing with etching and drying, samples were coated with platinum using vacuum sputtering (10 mA, 300 s) prior to SEM observation. The top and fractured surfaces of PPDO samples were delicately solvent etched, then dried at room temperature prior to sputter coating.

Synchrotron microbeam wide-angle X-ray diffraction (WAXD). Microbeam wide-angle X-ray scattering measurements were performed at the beamline TPS-25A of the National Synchrotron Radiation Research Center (NSRRC, Hsinchu, Taiwan) with an X-ray photon energy of 15 keV, a detector distance of 0.136 m, a wavelength of 0.8251 Å, and a q-range between 0.0629 and 3.53177 Å^−1^. An X-ray microbeam of 5 µm × 7 µm was directed to travel along a specified direction with 4 µm intervals, focusing on micro-spots of the PPDO ring-banded spherulites. One-dimensional profiles were used to analyze the specific micro-domains. Two-dimensional patterns were used to analyze the ridge and valley of lamellae orientation.

Synchrotron microbeam small-angle X-ray scattering (SAXS). Microbeam small-angle X-ray scattering measurements were performed at the beamline TPS-25A at the NSRRC with an X-ray photon energy of 15 keV, a detector distance of 5.09 m, a wavelength of 0.8251 Å, and a q-range between 0.00292 and 0.33545 Å^−1^. An X-ray microbeam of 5 µm × 7 µm was directed to focus on pre-determined micro-spots of the PPDO ring-banded spherulites. The X-ray 2D scattering patterns were used to analyze the PPDO crystal assembly. The lamellar parameters, specifically the long-period (Lo), crystal lamellar thickness (Lc), and amorphous lamellar thickness (La), were determined utilizing the 1D correlation function.

## 3. Results and Discussion

Past investigations available in the literature on PPDO mostly addressed 2D morphology on surface topology in thin cast-film specimens, and missed the interior views in constructing justifiable assembly mechanisms [16]. It was not until 2020 that K.Y. Huang et al. [36] and Y.J. Huang et al. [33] probed the 3D interior morphology of neat PPDO and PPDO/PESu, respectively, to shed new light on assembly mechanisms. On this basis, this study probed further, using synchrotron X-ray micro-diffraction to analyze the detailed nanoscale assembly. The assembly of lamellae in melt-crystallized neat PPDO is naturally quite compact and jammed, which has made it difficult to view the interior. Therefore, this study attempted to blend PVA with PPDO in order to loosen up the interior lamellae of PPDO for convenience of analysis. 

Figure 1a–c shows POM images that reveal the ring patterns of PPDO/PVA (95/5) crystallized at Tc = 35, 45, and 55 °C, respectively. With an increase of Tc from 35 °C to 70 °C, the spherulite size increases. For brevity, POM images in a wider range of Tc are placed in ESI Appendix A. More critically, the fractal growth of cactus-arm-like branches becomes more apparent, where periodic bands are visible in each of the spoked branches. That is, the bands are not smoothly circular around a common nucleus center, but are discretely and discontinuously divided/sectored on each of the “spoked arms” of the dendrite-like patterns. The inter-band spacing, however, does not differ much with the increase of Tc. Such spoke-divided ring bands in discrete and separate arms become increasingly striking at elevated Tc such as 55 °C, and are illustrated at the central core PPDO spherulite in Figure 1c. A drawn scheme is inserted at the core to depict the spoked ring bands.

With a slight increase in the PVA content of the PPDO/PVA blend, the interfaces in the cactus-arm-like dendrites become more obvious. Figure 2 shows POM images revealing the ring patterns of PPDO/PVA (90/10) at two Tc (45 and 50 °C), respectively. The dendrite-like branches become more apparent, and fractal-branching fibrils are aligned periodically to form successive ring bands on each of the spoked arms. For simplicity, this study focused on morphological analysis of the bands of PPDO/PVA (95/5) crystallized at Tc = 65 °C that gave the most regular banding patterns on the arms (spokes).

Solvent etching was necessary to expose the interior assembly. The PPDO/PVA (95/5) specimens were etched using p-dioxane prior to SEM interior analysis. At ambient temperatures, PVA rapidly dissolves in p-dioxane; by contrast, PPDO needs to be heated to dissolve in the same solvent. Thus, the differential solubility of these two polymers in p-dioxane has made it ideal for etching PVA from crystallized PPDO/PVA spherulites. Specimens were pre-etched and then characterized using SEM, the results of which are shown in Figure 2. The interfaces between the spokes became more apparent, with fibril lamellae clearly exposed (Figure 2a,b). Figure 3c shows a zoomed-in image of the red-squared zone in Figure 3b, where the ring bands are composed of cyclically alternate circumferential-oriented flat-on plates and radial-oriented edge-on crystal plates that bend and scroll in the clockwise direction to eventually merge into the circumferential-oriented flat-on plates. The thickness of these edge-on crystal plates is estimated to be ca. a few hundred nanometers, with a length of ~5–10 µm (i.e., equal to the inter-band spacing). Cycles repeat in the same manner to build successive bands in each of the spokes. Obviously, the growth is in a cycle and not continuous, unlike a screw helix radiating out from a common nucleus center. Instead, discontinuity between successive bands is obvious. Furthermore, the bands discretely self-assemble in each of the radial-oriented spokes (cactus-like arms), that is, they are not circularly smooth around a common nucleus.

Figure 4a,b show stepwise zoom-ins on the ring bands along a single spoke. A closer inspection reveals that the ridge zone is populated with numerous crevices, which are interfaced by edge-on crystal plates. By contrast, the circumferential-oriented valley zone, packed with stagger-overlapped flat-on plates, displays no crevices at all. Figure 4c is a line drawing showing that the spoked bands are evidently present on each of the cactus-like arms evolving as fractal branches originally from a nucleus center. Figure 4d shows that the edge-on plates are likely branches evolving from the circumferential flat-on parental crystals on the circumferential-oriented “valley”. The edge-on branches are at an oblique angle of ca. 25–45° from the circumferential-oriented and flat-on parental crystals. The initial oblique angle is ca. 25° or no greater than 45°; certainly, with gradual bending of the edges of branches to merge into the next valley zone, the oblique angle may increase from the original 25° to 60° or 90° angles. Such growth with oblique branching angles is typical for most polycrystalline crystals with branching characteristics.

The discussion now covers 3D interior dissection, where crystallized PPDO/PVA (95/5) specimens were fractured in both radial and circumferential directions. Figure 5a shows SEM micrographs of the PPDO spherulite fractured radially to expose the interiors. Figure 5b is a magnified zoom-in of the yellow-squared zone in Figure 5a, which covers both the top surface and fractured interiors. Figure 5c,d illustrates that the periodically banded PPDO/PVA (95/5), fractured along the radial direction, displays clear correlation between the top-surface ridge-to-valley assembly vs. the interior vertical-to-horizontal lamellae.

Circumferential-fractured interiors of the banded PPDO were then analyzed. Figure 6 shows SEM micrographs fractured along the circumferential direction of a specific banded PPDO. In principle, along the circumferential direction, fracturing can statistically cut all on ridge zones (Figure 6a), or alternatively on a ridge-to-valley zone (Figure 6b). In Figure 6a, one can see that if the fracture line is only on the ridges, the exposed interior lamellae are all normal-oriented (normal to the substrate). By contrast, if the fracture line covers both the ridge and valley zones, the interior lamellae underneath a ridge are still normal-oriented; yet, if across a valley zone, the lamellae underneath the valley zone are obviously horizontal-oriented. The fracture line in Figure 6b is drawn successively across two ridge zones and two valley zones; consistently, the alternate interior lamellae orientations are correlated well with the top surface ridge and valley. The interior dissection results of the periodically banded PPDO clearly do not support the conventional notion that the interior lamellae undergo a continuous helix twist to form optical bands in POM, as proposed in decades of classical investigations.

In the context of the PPDO/PVA (95/5) system, a thorough investigation was carried out by mapping the real-time micrograph shown in Figure 7a. An 80 × 40 µm block was delineated, encompassing the left hemisphere of the spherical core. A point-by-point scan was systematically performed within this designated region, with step-moving points spaced at 4 µm intervals. At each scan position, the synchrotron X-ray facility setup and software allowed for simultaneous WAXD/SAXD data to be acquired, and the resulting signals were amalgamated into a mapping profile for more precise analysis. The lamellar alignment within the diffraction results was consistent with those obtained from the preceding SAXS experiment, thereby reinforcing the universal robustness of the findings. Specifically, when the internal lamella plate is positioned at the ridge, it exhibits a perpendicular orientation to the substrate, resulting in a heightened signal indicated in red/yellow (Figure 7b). Conversely, when the internal crystal plates are positioned at the valley, aligning themselves parallel to the substrate, this orientation gives rise to signal deficiency, as portrayed in purple in the figure. This comprehensive analysis serves to corroborate the periodically ring-banded morphology observed through POM, which further affirms the accuracy of the internal crystal plate arrangement via SEM interior dissection.

Subsequently, six key spots were selected along the radial axis for further in-depth analysis, as illustrated in Figure 7b. Among these, spots #1, #2, #5 and #6 corresponded to the ridge zone, while spots #3 and #4 represented a zone in the valleys. The signals gathered from these six points were systematically organized into a 1D plot and presented in supporting information Appendix A. To further affirm the spatial position of the crystal planes, the 2D patterns were scrutinized, as demonstrated in Figure 7c. The discernment of ridge or valley positions from the SAXS signals in close proximity to the beam stop indicated the presence of a SAXS signal for the ridge and the absence for the valley. Observations revealed the presence of the (210) signal at both the ridges and valleys, while the (020) signal exclusively manifested only in the valleys. The crystal plane (020) along the *ab*-axis confirmed the horizontal orientation of crystals in the valley region at spot #3 and #4. The vertical orientation of the lamellae in the ridge region was substantiated by the presence of both SAXS and WAXD (210) crystal signals on spots #1, #2, #5, and #6.

Further, a more comprehensive analysis by synchrotron microbeam WAXD was undertaken to scrutinize the signal disparities between the ridge and valley zones, as depicted in Figure 8. As the microfocal beamline intersected with the ridge, the internal crystal plate adopted a vertical orientation to the substrate, assuming a normal configuration. In this scenario, the *b*-axis of the orthorhombic cell aligned in parallel to the beamline, and likewise, the plane (210) also paralleled with the beamline. This alignment resulted in the conspicuous generation of a signal, while the plane (020) remained perpendicular to the beamline, precluding any signal detection. Conversely, as the beamline targeted the valley region, as characterized by a horizontal orientation of the crystal plates, the *c*-axis of the orthorhombic cell aligned parallel to the beamline. In this case, both crystal planes (210) and (020) assumed parallel orientation to the beamline, thus leading to the generation of signals for both planes. By drawing a summary from the aforementioned results, evident variations in the arrangement of the crystal plates can be emulated. This simulation lends support to the internal crystal-plate arrangement as observed under SEM analysis, conclusively affirming the discontinuous grid-like nature of the internal lamellar arrangement. Consequently, achieving a precise morphological comparison with the SEM morphology necessitates the use of a dedicated SAXS microbeam characterization.

The dedicated SAXS microbeam measurements of PPDO/PVA were taken, and compared with SEM analysis in Figure 9. In the PPDO/PVA (95/5) system, the beamline was moved along the radial direction form spot #1 to #4 across several successive rings. It was aligned on the ridge and valley, and the corresponding positions on the POM and SEM were marked, as depicted in Figure 9a,b. The white circles in the figures indicate the relationship between the actual size of the beamline and the spherulite. Spots #1 and #3 correspond to the SAXS signals from the valley, while spots #2 and #4 correspond to the signals from the ridge. These four spots in the figure correspond to 2D-SAXS patterns, as given in Figure 9c. The microbeam used in this study has a size of approximately 6 µm. The distance between the rings formed by this mixing system at Tc = 65 °C is roughly 20 µm, which is about three times the size of the microbeam. Consequently, this gap is large enough to allow for further observation and analysis of specific positional variation on the successive rings.

When the X-ray microfocal beamline struck the elevated ridge on the upper surface, the internal crystal plates were arranged perpendicular to the substrate, causing the beamline to align parallel to the internal crystal plates and resulting in an evident SAXS signal. Conversely, when the beamline struck the valley on the upper surface, the internal crystal plates were arranged parallel to the substrate, causing the microfocal beamline to be perpendicular to the internal crystal plates, resulting in no signal at all. The discrepancy between these two opposite signals confirms the alternate and mutual perpendicular arrangements of the internal crystal plates. It is noteworthy that the 2D SAXS diagram of this system exhibits a different arc shape, with angles between the central lines measuring approximately 25–45°, as shown in Figure 9d. The observed SAXS signal provides evidence to suggest that within the ridges, there may exist multiple orientations of tens of stacks of lamellae, and thus the SAXS signal tends to be a statistical average of these lamellae of varying orientations. This phenomenon points to a potential structural complexity within the system, where the lamellae are not uniformly oriented upon crystallization from molten melt, but rather exhibit variations in their alignment due to crowded compactness and impingement.

Furthermore, the calculated lamellar parameters (L_o_ = 8.1 nm, L_c_ = 4.75 nm, and L_a_ = 3.35 nm) of the SAXS signals, determined using the 1D correlation function (as shown in ESI Appendix A), provide significant insights into the structural attributes of the lamellae. The SEM image presented earlier in Figure 4 provides a valuable observation on the top-surface morphology of the etched sample. Notably, it shows that the vertical lamellae situated on the ridges exhibit a complex fractal structure with multiple branches. These side branches are observed to incline from the main lamellar stem at an angle of approximately 25–45° (varying slightly with changes in inclination). This branching pattern adds an intriguing level of intricacy to the material’s microstructure, suggesting a dynamic interplay of stresses or growth mechanisms that contribute to this unique lamellar arrangement. Moreover, when comparing the SEM and SAXS results, it becomes evident that these branching phenomena form arc-shaped patterns, further highlighting the correlation between structural features observed at different length scales.

Structural coloration by interference with white light is commonly seen in bio-species that have developed orderly grating assemblies for iridescent coloration through millions of years of evolution. In previous studies [37], we have reported iridescence in ring-banded polymers. Liao et al. [35] proved that ring-banded PHB spherulites, with orderly parallel crystal plates in periodicity, produced similar functions. Observation was also extended to other polymers. Nagarajan et al. [37] proved that ring-banded PBA spherulites are packed with orderly-aligned interior and top-surface crystal plates for producing iridescence similar to those seen in naturally evolved bio-species. Such features can function as structural color crystals, and can also support lamellae to assemble as orderly self-aligned gratings with nanostructures and microstructures that are suitable for, or at least capable of, interferences with the white light of sub-micrometer wavelengths between 400–800 nm. 

Iridescence tests similar to those reported in the literature [35,37] were performed on specimens of PPDO/PVA (95/5) crystallized at Tc = 45 °C to 70 °C (at 5 °C intervals). Note that the ring patterns in PPDO/PVA (95/5) differ slightly from those in PBA or PHB, which display superficially smooth circular rings. PPDO/PVA (95/5) displays cactus-arm-like dendrites with ring bands on individual arms, and is thus discretely not continuous. By contrast, PBA and PHB have smoothly circular (or spiral) ring bands surrounding a common nucleus center, with better regularity and order in assembly. Nevertheless, the iridescence results in Figure 10 show that as long as the lamellae are assembled into orderly periodic ring bands in PPDO/PVA, they are potentially capable of displaying iridescence. The intensity of iridescence, however, is qualitatively less pronounced than that seen in PBA or PHB [35,37]. The banded PPDO at Tc = 45 °C displays weak but observable iridescence; as Tc is increased to 55–60 °C, where PPDO has the most regular dendritic bands, it exhibits the most pronounced iridescence. The banded PPDO at Tc = 70 °C, though still roughly banded, possesses inter-band spacing that is too large and thus unsuitable for effective interference to become iridescent, leading to nearly non-iridescence. 

## 4. Conclusions

PPDO is conventionally known to exhibit circular ring bands upon crystallization in a suitable range of Tc. This study has advanced to its microstructures and nanostructures, leading to periodicity of assembly in discrete sectors (cactus-like spokes) using microscopy and micro-beam X-ray analyses. However, upon introducing a diluent PVA capable of hydrogen bonding interactions with PPDO, the result is a cactus-arm-like morphology, where interfaces between the cactus-arm-like dendrites become obvious. This morphological fact suggests that the original smooth, circular ring bands in neat PPDO might be composed of spoke-like dendrites radiating outward from a nucleus center. The circumferential direction of the circular ring bands may not be continuous, appearing as closely jammed spokes that are assembled side by side. Upon crystallization with diluent PVA, the side-by-side spokes are detached to appear as cactus-arm dendrites with fractal branches. Ring bands appear on each of the discrete, fractal-grown arms of the detached spokes, leading to non-circular patterns as viewed in POM. The SEM analyses reveal that the ring bands are composed of cyclically alternate circumferential-oriented flat-on plates and radial-oriented edge-on crystal plates that bend and scroll in the clockwise direction to eventually merge into the circumferential-oriented flat-on plates. The thickness of these edge-on crystal plates is estimated to be ca. a few hundred nanometers, with a length of ~5–10 µm (i.e., equal to the inter-band spacing). Cycles repeat in the same manner to build successive bands in each of the spokes.

To further support the results of microscopy dissection, synchrotron radiation microfocal beams of both small-angle X-ray scattering and wide-angle X-ray diffraction were used to analyze the periodic crystal aggregates on specific spots of the unusual PPDO dendritic ring bands. The critical microbeam 2D SAXS and WAXD analyses further justify and strengthen the fact that the microscopic lamellar assembly in the PPDO banded periodic architectures is packed with alternately normal-oriented strut-crystals periodically interfaced with horizontal-oriented rib fibrils. The oblique angles between the flat-on strut lamellae with edge-on branches are ca. 25–45° (depending on gradient inclines) as visible in the 3D microscopy dissection, which is also proved by the azimuthal angle in SAXS. The grating-like strut-rib assembly in the PPDO cactus-arm-like ring bands mimic natural structural coloration crystals, and is further proved by iridescence tests.

## Figures and Tables

**Figure 1 nanomaterials-13-02665-f001:**
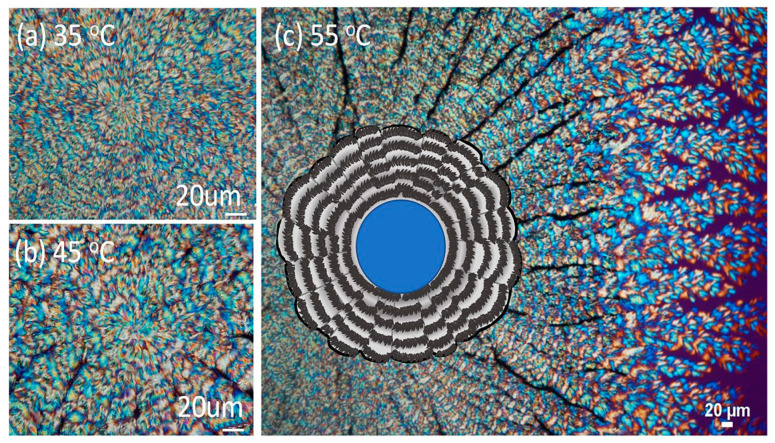
POM graphs of PPDO/PVA (95/5) blend crystallized with cover at various Tc: (**a**) 35 °C, (**b**) 45 °C, and (**c**) 55 °C, where a scheme is inserted to depict the spoked ring bands.

**Figure 2 nanomaterials-13-02665-f002:**
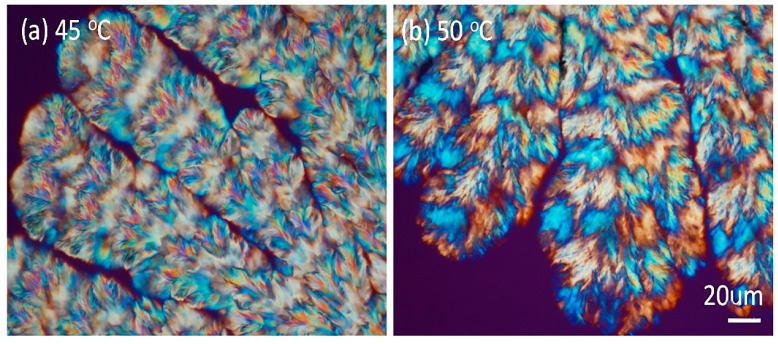
Dendritic cactus-like arms packed with individual periodic bands in PPDO/PVA (90/10) crystallized at Tc = 45 and 50 °C, respectively.

**Figure 3 nanomaterials-13-02665-f003:**
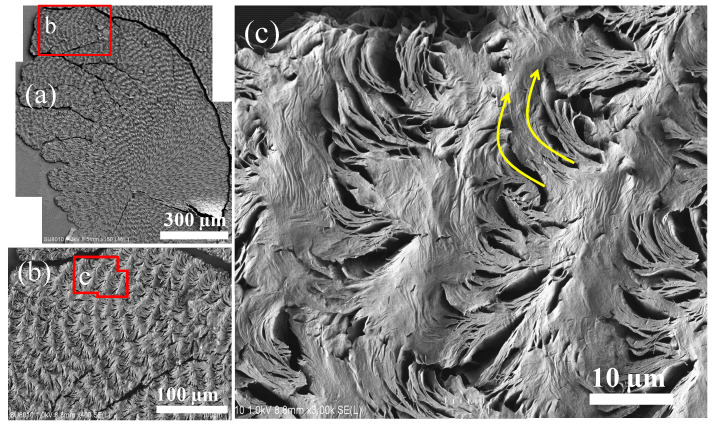
SEM graphs for the top surface of p-dioxane-etched PPDO/PVA (95/5) blend crystallized at Tc = 65 °C: (**a**) image of the whole spherulite, (**b**) zoomed-in areas in the red box (**a**), and (**c**) zoom-in of the red block in Graph (**b**), with yellow arrows revealing the lamellar bending direction.

**Figure 4 nanomaterials-13-02665-f004:**
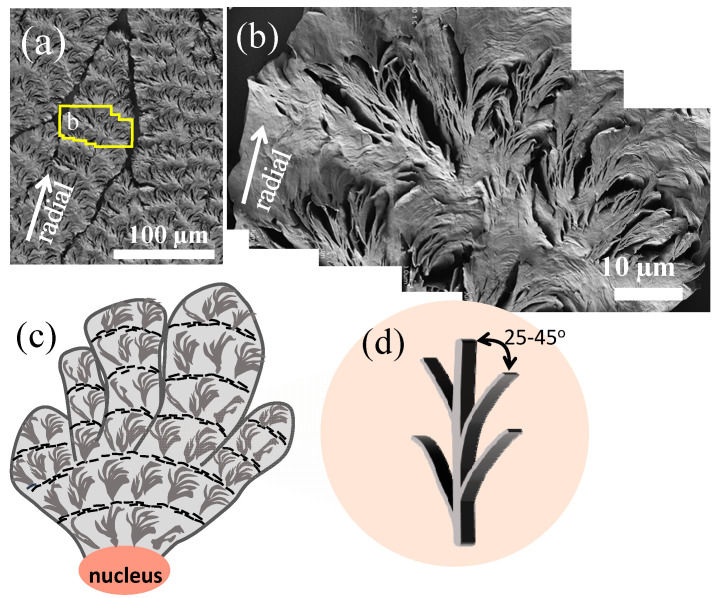
(**a**) SEM micrographs, (**b**) zoomed-in area in figure (**a**), and (**c**) scheme of the crystalline arrangement on the top surface of PPDO/PVA (95/5) at Tc = 65 °C (**d**) enlarged dendrite scheme.

**Figure 5 nanomaterials-13-02665-f005:**
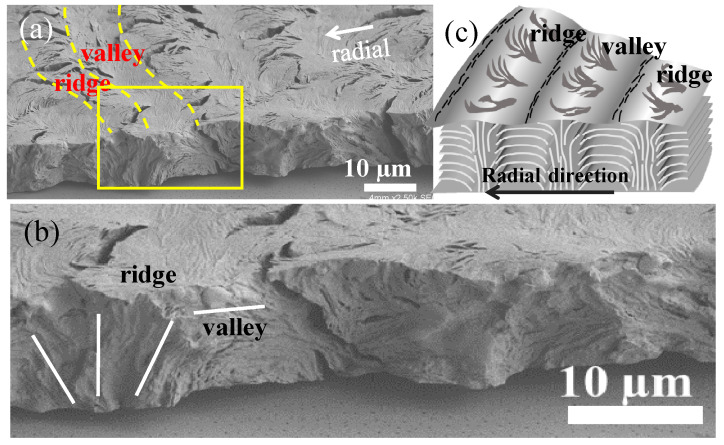
(**a**,**b**) SEM micrographs of the spherulite fractured radially and (**c**) schematic illustration of PPDO ring-banded spherulites from PPDO/PVA (95/5) at Tc = 65 °C.

**Figure 6 nanomaterials-13-02665-f006:**
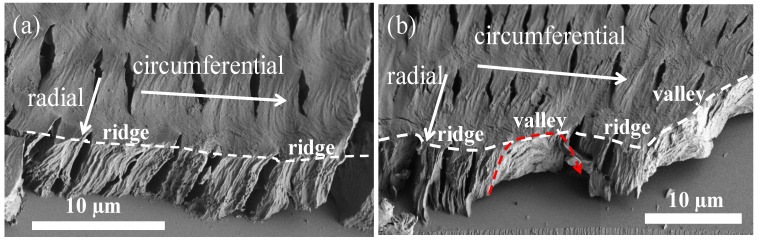
SEM micrographs of a PPDO banded spherulite crystallized from PPDO/PVA (95/5) at Tc = 65 °C, fractured tangentially (i.e., circumferentially) in two different zones: (**a**) across the ridge zones only, and (**b**) across the ridge–valley transition zone.

**Figure 7 nanomaterials-13-02665-f007:**
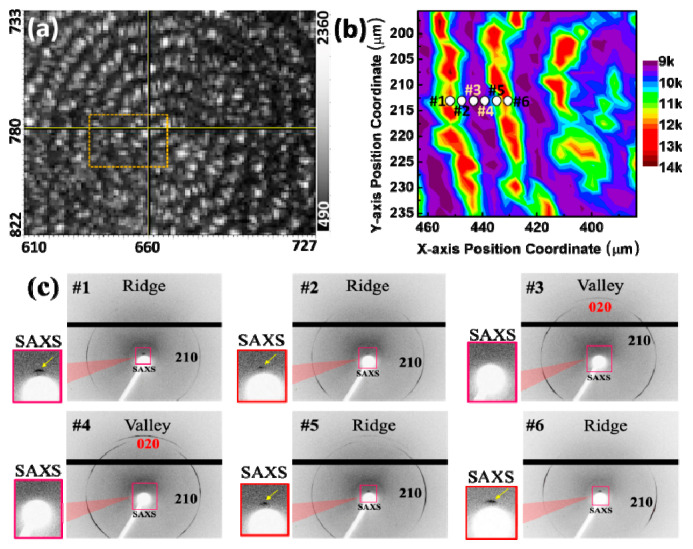
(**a**) Real-time OM micrograph of a banded PPDO spherulite with a yellow dotted box revealing the zone for simultaneous 2D-WAXS/SAXS microbeam measurements, taken at 4 µm intervals, (**b**) mapping profile of the SAXS signal, and (**c**) 2D patterns from micro-locations #1 through #6, depicting the valley zone (spots #3 and #4) and ridge zone (spots #1, #2, #5, and #6), respectively, with a microbeam size of 6 µm, PPDO/PVA (95/5) crystallized at Tc = 65 °C.

**Figure 8 nanomaterials-13-02665-f008:**
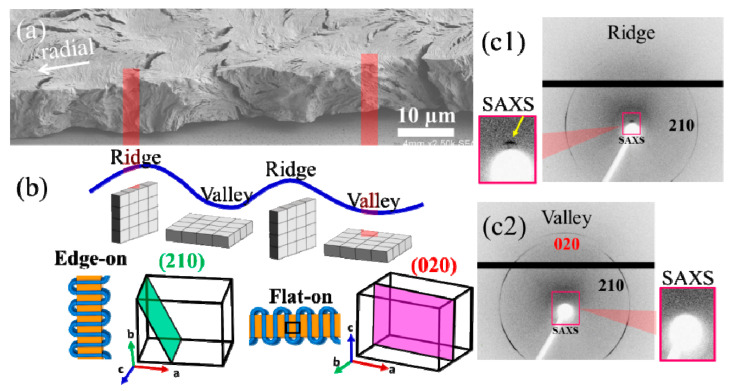
(**a**) SEM image, (**b**) scheme revealing lamellae/unit-cell orientation in ridge and valley, (**c1**,**c2**) synchrotron microbeam X-ray diffraction (WAXD and SAXS) in a PPDO/PVA (95/5) ring-banded spherulite at Tc = 65 °C.

**Figure 9 nanomaterials-13-02665-f009:**
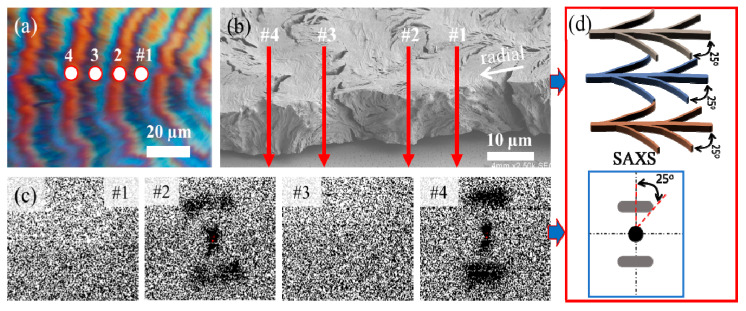
(**a**) POM graph, (**b**) SEM graph for radial-fractured interiors, (**c**) synchrotron microbeam SAXS 2D pattern on spots #1 to #4 of the PPDO/PVA (95/5) ring-banded spherulite at Tc = 65 °C [microbeam size = 6 µm], and (**d**) scheme of lamellae branching angles in the ridge zone as revealed from the corresponding 2D-SAXS.

**Figure 10 nanomaterials-13-02665-f010:**
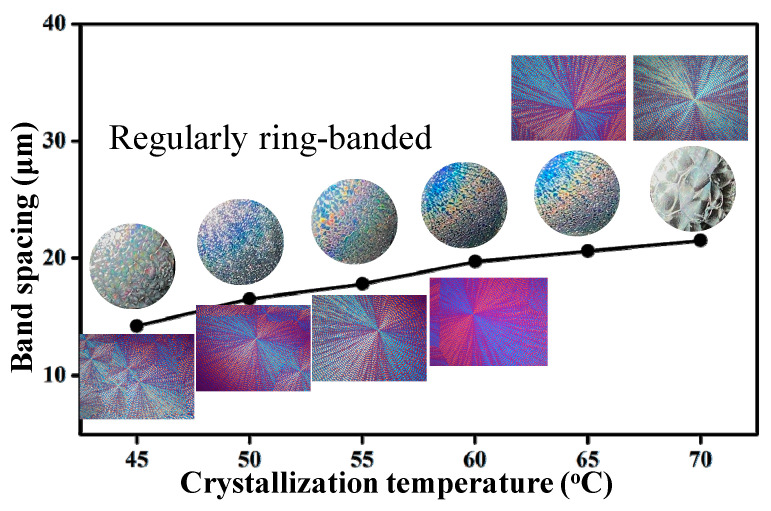
Summary comparison of photonic iridescence correlating with the ring patterns and inter-band spacing for PPDO/PVA (95/5) crystallized at Tc ranging from 45 °C to 70 °C.

## Data Availability

The data presented in this study are available on request from the corresponding author.

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
