# Peer review of "Probing the Nano-Assembly Leading to Periodic Gratings in Poly(p-dioxanone)"

_nanomaterials, 2023, doi:10.3390/nano13192665_

Round 1

Reviewer 1 Report

The temperature dependent and crystallization behaviour of poly(p-dioxanone), PPDO, has been studied in detail by the authors and is reported in this article. Methods used include synchrotron radiation supported SAXS and WAXS methods. The experiments are generally well described - the preparation of the samples should be repeatable by other groups. The overall presentation of the structural data (SAXS, WAXS) is excellent. As PPDO has gained quite some interest for several applications including the fabrication of medical devices I support the publication of this study in 'Nanomaterials'. However, the results are of course only of relevance for this very system under the exact same conditions. I expect that a  moderate change in parameters will have a significant impact on the morphology of the films prepared, the scope of the study is hence quite narrow. Nevertheless, the experimental data merit publication of the article after careful language revision. (among others: e.g page 3, top paragraph, '...preset at different crystallization temperature for fully crystallized. ...')

see above

Author Response

Please find it in the attchment

Reviewer 2 Report

The study entitled „Three-Dimensional Dissection with Microbeam X-ray Diffraction Probing the Nano-Assembly Leading to Periodic Gratings in Poly(p-dioxanone)” presents a comprehensive analysis of the morphology of poly(p-dioxanone)/poly(vinyl alcohol) crystal aggregates. In my opinion, the study is well-written, interesting and innovative. Special attention deserves the application of various advanced methods including synchrotron microbeam X-ray, high-resolution field-emission scanning electron microscopy and polarized-light optical microscopy.

I do not find any methodological shortcomings in the work. I fully agree with the discussion of the results and conclusions. Therefore, I believe the manuscript can be accepted in its current form.

Author Response

please find it in the attachement

Reviewer 3 Report

Authors analyze the crystal aggregates, in the form of cactus-arm-like ring bands, of PPDO and PVA blends through extensive microscopic experiments. The resulted morphologies are clearly present and definitely point to an assembled structure where specific angles are formed between the valley domains and the ones in branches. The work is interesting and well presented, as long as the experimental images are presented and explained. However, the manuscript requires revisions to increase its clarity and reading flow. The present contribution falls within the scope of Nanomaterials and subject to addressing the comments below could be publishable.

Authors should increase the clarity of their manuscript. For example, some figure legends are misleading like in Figure 3. Panel (d) is mentioned but such panel does not exist. According to what is presented panel (a) is the whole image, panel (b) zooms in in the read square of panel (a) and panel (c) zooms in the red area of panel (b). Then, two yellow arrows appear in panel (c) without any corresponding description of their functionality. Another example is Figure 9 where panel (a) is not mentioned in the legend (should proceed “POM graph”).

Another example of reduced clarity is the present title of the manuscript, as it is too long and rather uninformative.

Due to errors in English grammar and syntax the clarity of the manuscript is not optimal. Some examples:

Physical quantities should be properly defined upon being introduced, even if their physical correspondence is trivial. For example T_c (page 2), the lamella parameters (page 9) etc.

Some sentences are quite long and could be split into two. Examples: “Unlike other commodity … “ (page 1); Page 5: “Instead … nucleus”. The meaning of the second part of the sentence is unclear and should be rephrased.

The word “respectively” is not used in its correct context. For example in Page 6 “…. circumferential direction, respectively.”.

Page 1: “varieties of methods” -> “various methods”; Page 2: “long chain polymers” -> “long-chain polymers” Page 2: “light on revealing the interior” -> “light on the interior”; Page 4: “at several T_c” -> “at T_C”; Figure 1 legend: “scheme inserted” -> “scheme is inserted”; Page 5: “differential solubility” -> “different solubility”; Page 5: “cartoon dramatizing” –> “sketch depicting”; Page 7: “exclusively manifests only” -> “manifests only”; Page 9: “form spot #1” -> “from spot #1”; Page 9: “This phenomenon points” -> “This points”; Page 10: “25-45 degrees angle” -> “25-45 degrees”; “morphological fact” -> “morphological trend”

The English syntax and grammar should be improved to increase the clarity of the manuscript, which is currently the main problem of the present contribution.

Author Response

please find it in the attachment.
